# New Virus Diagnostic Approaches to Ensuring the Ongoing Plant Biosecurity of Aotearoa New Zealand

**DOI:** 10.3390/v15020418

**Published:** 2023-02-01

**Authors:** Catia Delmiglio, David W. Waite, Sonia T. Lilly, Juncong Yan, Candace E. Elliott, Julie Pattemore, Paul L. Guy, Jeremy R. Thompson

**Affiliations:** 1Plant Health and Environment Laboratory, Ministry for Primary Industries, P.O. Box 2095, Auckland 1140, New Zealand; 2Science and Surveillance Group, Post Entry Quarantine, Department of Agriculture, Fisheries and Forestry, Mickleham, VIC 3064, Australia; 3Department of Botany, University of Otago, P.O. Box 56, Dunedin 9054, New Zealand

**Keywords:** Aotearoa, Māori, Oxford Nanopore, Illumina, Mickleham, post-entry quarantine, point-of-use, environmental RNA/DNA

## Abstract

To protect New Zealand’s unique ecosystems and primary industries, imported plant materials must be constantly monitored at the border for high-threat pathogens. Techniques adopted for this purpose must be robust, accurate, rapid, and sufficiently agile to respond to new and emerging threats. Polymerase chain reaction (PCR), especially real-time PCR, remains an essential diagnostic tool but it is now being complemented by high-throughput sequencing using both Oxford Nanopore and Illumina technologies, allowing unbiased screening of whole populations. The demand for and value of Point-of-Use (PoU) technologies, which allow for in situ screening, are also increasing. Isothermal PoU molecular diagnostics based on recombinase polymerase amplification (RPA) and loop-mediated amplification (LAMP) do not require expensive equipment and can reach PCR-comparable levels of sensitivity. Recent advances in PoU technologies offer opportunities for increased specificity, accuracy, and sensitivities which makes them suitable for wider utilization by frontline or border staff. National and international activities and initiatives are adopted to improve both the plant virus biosecurity infrastructure and the integration, development, and harmonization of new virus diagnostic technologies.

## 1. Background—New Zealand’s Uniqueness

New Zealand (NZ), Aotearoa in Māori, is a relatively young country in terms of human history, with the ancestors of the Māori arriving from Polynesia probably between 1200 and 1300 AD but remaining relatively sparsely populated until European settlements in the 1800s [1]. Its long isolation from other land masses and its diverse geography and climate have resulted in the development of unique flora and fauna. With an estimated 80,000 native species [2], NZ is an internationally recognised global ‘hotspot’ for biodiversity with 81% of insects, 84% of vascular plants, 88% of freshwater fish, 72% of birds, and 100% of reptiles, frogs, and bats endemic to NZ (indigenous species found nowhere else) [3,4]. Sustained geographic isolation has kept out many pests and diseases but, with the increased movement of people and goods, NZ is vulnerable to invasion by exotic species, and consequently the number of endemic species has been in constant decline since settlement. Changes in land use, habitat loss, and the introduction of exotic plants, animals, and diseases that hitchhiked with the introduced plants and animals have all contributed to the extinction of at least 80 species from the total biota, and an increasing number of species are considered threatened or critically endangered [5].

NZ is also unique for its relationship with its indigenous people, Māori, the government’s formal partner since the signing of the Treaty of Waitangi (Te Tiriti o Waitangi) in 1840 [6,7]. Māori have an intimate relationship with the natural environment, one of guardianship, stewardship, and protection [8], and an inclusive collaborative approach with Māori in national and local biosecurity strategies can provide efficiencies for the sustainable management of the land and resources, as well as meeting indigenous well-being aspirations [6]. Globally invasive species, including viruses, threaten not only a country’s biosecurity and economic trade, but also indigenous cultural practices and economic activities because of their potential impact on native species [6,8].

### 1.1. Exotic Invasions

Introduced plants have played a significant role in NZ’s economy and culture since the arrival of Polynesians in the 13th century, but it was the arrival of Europeans in the 1800s that led to a drastic increase in the rate of plant introductions [9]. NZ has the second-highest recorded number of invasive plant species in the world, excluding overseas territories [5,9], with over half of all gymnosperms and angiosperms in the country being exotics [10]. NZ’s environment is highly suited to primary production, with the agricultural sector consistently accounting for 9–10% of the annual total gross domestic product (GDP) since 2016 [11]. NZ has also gained an international reputation as a source of quality produce. Five markets dominated 67% of NZ’s total horticultural exports for the year to March 2022 (Continental Europe, China, the United States of America [USA], Japan, and Australia)—with kiwifruit, wine grape, apples, and pears as the dominant export crops by value [11]. Lesser-recognised crops such as avocado and hops have increased in export value in the past ten years—avocado fruit and oil exports increased from NZD62 million (m) in 2011 to NZD211 m in 2021, and hops from an estimated NZD9 m to NZD36 m for the same period [12]. NZ also imports fruit, vegetables, grains, and seeds mostly from Australia, USA, Ecuador (for bananas), and smaller amounts from continental European and Asian countries (Figure 1). Expansion of horticultural production requires that growers continually source new germplasm from overseas, and the import of fresh produce, cut flowers, nursery stock, seeds, and grain creates pathways for the potential entry of exotic plant diseases, including viruses.

Over the past two centuries exotic plants have established in NZ, bringing with them some of their viruses and many of their invertebrate vectors [13,14]. Records of plant viruses increased from around 100 species in the 1990s [14,15,16,17] to 258 species in 2021 [18] (Figure 2). Most plant virus records in NZ are from economically important plants including horticultural, ornamental, and unwanted invasive plant species (the latter acting as virus reservoirs) [14,16,18]; however, it is probable that some of these new viruses are endemic to NZ [14,18,19]. The increase in the cumulative total number of viruses reflects not only new incursions but also changing technologies and our ability to detect and identify viruses. A large proportion of the viruses recorded are aphid-transmitted: 54 potyviruses, 26 carlaviruses, and 15 luteoviruses make up over one-third (37%) of the records. Except for the thrips-transmitted tomato spotted wilt virus (48 species), the viruses recorded from the majority of hosts are also aphid-transmitted: cucumber mosaic virus (93 species), alfalfa mosaic virus (58 species), turnip yellows virus (39 species), and watermelon mosaic virus (32 species). A recent review by Guy et al. [14] identified that at least 16 viruses had infected native plant species (including native grasses, cresses, solanums, and cucurbits) likely jumping from crops, weeds, and garden ornamentals.

The use of high-throughput sequencing (HTS) in plant diagnostics has accelerated the discovery of new viruses in native plants in recent years [18,19,20], including potential endemic viruses from the *Emaravirus* genus [18], such as the recently identified and characterized Karaka Ōkahu purepure virus [19]. There are still limited data available on the true extent of virus diseases in native ecosystems in NZ with information available from only five of the ten largest plant families [14]. From the current data it is possible to see that of the introduced viruses, those that are aphid-transmitted have the highest percentage (40%) with luteoviruses, carlaviruses, and potyviruses as the main groups [14,16]. None of the endemic aphid species are associated with virus transmission [14]; the vectors of invasive viruses are therefore most likely to be introduced polyphagous species with broad host ranges. Native ecosystems have been almost entirely replaced with highly modified ecosystems where exotic species dominate [21]. The presence of virus-infected exotic species has been shown in studies [22,23] to result in a higher incidence of viruses in co-localised native species [14].

Changes in land usage and intensification, with changes in NZ’s land cover, are driving more habitat fragmentation that allows for the spread of invasive species, a process exacerbated by climate change. In NZ, the annual average temperature increased by 1.13 °C (± 0.27) over the period 1909 to 2019, and 28 out of 30 monitored sites showed an increase in annual average temperature for the 1972–2019 period [5]. Changes in temperature can affect weather patterns, and climate change has contributed to 12 extreme rainfall events in NZ between 2007 and 2017 and two major drought events in 2007–2008 and 2012–2013 [5].

### 1.2. New Zealand Biosecurity

NZ has a multi-layered biosecurity system that aims to exclude, eradicate, or manage pests and pathogens that pose a risk to the NZ economy, environment, and way of life. The system begins offshore, implements protective measures at our borders, and continues post-border inside NZ [24]. At each of these stages systems, policies, regulations, and practices are in place to prevent pests and pathogens from entering NZ and to prepare for and respond to any pest incursions that may occur. Contingency plans for eradicating pests, or for managing the impact of incursions, have been developed and successfully employed in the past. Recent examples include the successful eradication of the Queensland fruit fly (*Bacterocera tryoni*) in 2015 and 2019 (see https://www.mpi.govt.nz/biosecurity/major-pest-and-disease-threats/queensland-fruit-fly/, accessed on 2 June 2022); and the management of the potato mop-top virus (PMTV) following the first incursion response in 2018 under a cost-share framework between industry and government. This management strategy has subsequently minimised any impact to the potato industry as shown by the low incidence of the virus in both the 2020/2021 and 2021/2022 seed potato crops (Iain Kirkwood, Potatoes New Zealand, pers. comms. 2 June 2022).

Post-border monitoring in NZ requires engagement and a joint effort by central government, regional councils, and industry, as well as local communities and Māori. The Ministry for Primary Industries (MPI) has the lead role for biosecurity in NZ, and the Plant Health and Environment Laboratory (PHEL) is the national reference laboratory responsible for the identification of exotic pests and diseases affecting terrestrial and freshwater plants and the environment. PHEL is an International Accreditation New Zealand (IANZ)-accredited laboratory that provides scientific advice and testing services for pathogens and invertebrate pests in both domestic and imported plant material. The facility also possesses a Level 3B Biosafety Containment greenhouse facility to provide post-entry quarantine (PEQ) for imported nursery stock. The PHEL virology team specialises in diagnostics of viruses and viroids which it achieves by continuously reviewing and developing diagnostic technologies and systems to meet the demands posed by emerging diseases in order to safeguard NZ’s biosecurity while responding to industry’s demands.

In the technical review that follows, it is described how, based on the uniqueness of NZ, these technologies will strengthen NZ’s plant biosecurity capability in light of predicted future challenges such as climate change and disease emergence, to name but two.

## 2. New and Old Technologies

The adoption of new technologies is frequently stalled by a natural reluctance to abandon the familiar in favour of the new. A case in point is the enzyme-linked immunosorbent assay (ELISA) which, despite well over forty years of service to the scientific community and the regular criticism of its comparatively low sensitivity (when compared to molecular tests), is still considered the workhorse of the virus diagnostician’s toolkit [25]. In adopting any new methodology Boonham et al. [26] caution that not only do the benefits of such a change need to be clearly demonstrated but also a procedure that facilitates initiation and ongoing use of that new methodology must be established. Often it is this latter requirement that creates a perceived lag in diagnostic laboratory capability.

At present the vast majority of validated and accredited diagnostic tests for plant viruses and viroids on germplasm entering NZ are polymerase chain reaction (PCR) assays—PHEL has specific and generic PCR tests validated to detect more than 400 viruses and viroids, and all 17 IANZ-accredited tests are PCR-based. Testing primarily consists of real-time PCR, followed up with a confirmatory end-point PCR and Sanger sequencing to verify positive real-time PCR results, where needed. The fact that close to forty years since its invention, PCR was the gold standard for high-throughput diagnostics during the COVID-19 pandemic [27] attests to its enduring power as a diagnostic tool. It is therefore likely to remain the go-to technique for maintaining PHEL’s virus diagnostic capacity, albeit one that is increasingly supplemented with new, complementary technologies. Recently massively parallel sequencing or High-Throughput Sequencing (HTS) has revolutionised the a priori identification of viruses and other organisms. Another technique which does not rely on prior sequence knowledge is mass spectrometry which, in contrast to other methods, can derive sequences from protein. This technique is under-utilised [28,29,30] in plant virus detection and characterisation. Below, future and existing activities to upgrade our national border diagnostics capacity and capabilities by harnessing the benefits of new technologies are discussed.

### 2.1. Oxford Nanopore Technologies

Oxford Nanopore Technologies sequencing has become a staple of the PHEL virology diagnostic tool kit, with 103 symptomatic samples obtained through domestic surveillance and post-entry quarantine analysed using the MinION Flongle platform since the beginning of 2021. The use of the MinION Flongle to detect a broad spectrum of deoxyribonucleic acid (DNA) and ribonucleic acid (RNA) plant viruses and viroids was enabled by the development of an in-house method for producing cDNA libraries from RNA transcripts that lack polyadenylated tails [31]. Multiple research projects are currently running to further operationalise this technology and expand its use into additional work streams, particularly where greater sensitivity is required, such as for screening asymptomatic plant material. There is continuous streamlining of the bioinformatic analysis of the sequence data produced through this technique to improve the turnaround time for sample screening while retaining sensitivity and accuracy and applying workflow management software such as Nextflow (https://www.nextflow.io/, accessed on 15 November 2022) to improve reproducibility of analyses. The ability to perform generic testing of samples, as opposed to the target-specific nature of PCR assays or ELISA, has allowed us to make several detections of virus-like sequences in previously unknown hosts [31] and to establish sequencing capacity for the increased demand for MinION-based sequencing.

While the current implementation of MinION sequencing is sufficient for the requirements today, there is a continuous search for new techniques made with this fast-developing technology to find new avenues to apply MinION sequencing. For example, the recently released adaptive read sampling technology provides a powerful means of reducing host detection in a sequencing run and therefore improves sensitivity. However, the method of cDNA generation from non-polyadenylated RNA yields libraries with a low n50 (typically less than 1000 bp) and assessments of the effect of read length on adaptive sampling demonstrate poor enrichment for sequences less than 5 kbp in length [32]. This is therefore an area that should be developed in order to improve the detection methodology, achieving an increase in sensitivity through improvements in host depletion or targeted enrichment of regulated viruses during sequencing.

There is also the recent release of short fragment sequencing capability in the MinKNOW platform (available in version 22.03) that would allow the sequencing of small RNA molecules, which are a desirable target for certain plant virus detections [33,34,35]. Methods that can enrich viral nucleic acids, either natively (such as small RNA sequencing) or through techniques such as PCR tiling and bead capture, are areas of active research. Applying such approaches will not only allow us to improve our diagnostic sensitivity, which is critical when applying HTS to asymptomatic plant material, but may also mitigate the need to perform post hoc validation of detections made through sequencing alone.

### 2.2. Illumina

High-throughput short-read sequencing is also a vital part of PHEL’s HTS capability that has been applied many times to yield viral genomes from infected plant material [36,37,38,39]. While Illumina platforms do not provide the rapid turnaround times of the Oxford Nanopore platform, they provide unparalleled depth of sequencing and quality which can be critical when attempting to recover rare viral sequences from diagnostic samples. Like the MinION platform, Illumina sequencing allows us to make detections in samples without prior knowledge of the infecting species, but the greater quality and sequencing depth allows for more confidence when the detection result is negative in asymptomatic samples.

Although the standard approach of deeply sequencing a sample and then screening for viral sequences is performed regularly, there are also investigations about new applications of the Illumina platform to expand the diagnostic capability. Of particular interest is the ability to find alternate lines of evidence of infection, rather than simply sequencing deeper and deeper. One such application could be as a means of detecting abnormal methylation events in the host genome, which can occur in response to viral infection [40,41,42,43,44,45], when plants may be asymptomatic or suspected to be carrying a low viral load. Methylation detection with Illumina sequencing is dependent upon bisulfite conversion of unmethylated cytosine, but 6 mA methylation events can still be indirectly captured using antibody capture and enrichment techniques prior to sequencing [45]. Enrichment of methylated regions can be performed prior to sequencing to increase the relative depth of coverage for targets of interest when the host immune response is causing the methylation of circulating DNA [42] or RNA [44] viruses.

In situations where symptoms abnormal for a given pathogen are encountered, there is an increasingly need to examine the genome of the detected pathogens to look for mutations such as single nucleotide polymorphisms and insertion/deletion events which might be associated with unexpected disease symptoms. It is possible that as the understanding of virology grows, it will be required to examine viral genomes in new ways, such as considering the influence of viral RNA secondary structures on gene translation [46,47] or expression levels [47,48,49] in infected plant material. The SHAPE-MaP [50,51,52] technique combined with Illumina sequencing may be used in the future to gain deeper insights into the functionality of pathogenic genomes generated as part of testing.

### 2.3. Point-of-Use (PoU)

Diagnostics are being revolutionised by HTS technologies due to the quantity and quality of the data they generate. Nevertheless, there is also a clear value and demand for diagnostic tools that allow for in situ quick screening by scientifically untrained operators. This is becoming an essential component in the fight against the spread of infectious diseases [53]. Not only does it provide a “frontline” during epidemics but importantly it shifts some of the diagnostic responsibility to the affected, whether they be patients, doctors, or farmers, thereby facilitating engagement, ownership, and a “demystification” of scientific methods. The COVID-19 pandemic especially has generated a new global interest in viruses and molecular biology and showcased the value of rapid PoU diagnostics [54].

The adoption and application of most novel PoU technologies in plant virology has lagged behind uptake in animal virology, with the notable exception of lateral flow immunoassays (LFIA), which have been in the hands of growers, extension specialists, and scientists for decades [55]. A number of PoU testing kits using recombinase polymerase amplification (RPA) technology both in end-point and real-time formats are now commercially available for plant pathogen diagnostics (e.g., from Agdia^®^). Like most PoU methodologies RPA reactions are isothermal and therefore do not require expensive equipment such as thermocyclers, but unlike LFIA they can reach PCR-comparable levels of sensitivity as demonstrated by the newly developed tomato yellow leaf curl virus and tomato spotted wilt virus assays [56,57]. A lateral flow readout has been added to RPA systems for plant virus detection, whereby labelled amplicons bind to membrane-immobilised complementary probes [57,58,59]. This adaptation has also been integrated into another commonly used isothermal detection, loop mediated amplification (LAMP), for the detection of both RNA and DNA plant viruses [60,61,62].

Further customisations of RPA and LAMP are underway with CRISPR-Cas chemistries, such as ‘specific high-sensitivity enzymatic reporter unlocking’ (SHERLOCK) [63] and ‘DNA endonuclease-targeted CRISPR trans reporter’ (DETECTR) [64] and their derivatives. Both methods exploit the collateral activity of Cas12 and Cas13 proteins to cleave DNA and RNA targets, respectively, through specially designed guide RNAs. They have already been recently used to detect both human viruses [65] and a number of plant RNA viruses [66,67].

Alongside advances in amplification-based detection technologies are developments in novel ligands, a new generation of synthetic “antibodies” that promise better molecular sensing and detection. Nanobodies, antibody fragments consisting of a single monomeric variable antibody domain [68], possess many advantages over traditional antibodies due to their higher target affinity and specificity, higher stability, and, due to their smaller size, higher solubility. They have recently been developed for the detection of Mal de Río Cuarto virus [69] and tomato leaf curl Sudan virus [70]. Another technology being considered is aptamers, single-stranded oligonucleotides that fold into three-dimensional structures which can bind with high affinity to a range of molecules and biomarkers. Their benefit over traditional antibodies is their simplicity, relatively low cost, and ability to be selected in vitro [71]. Their use for plant virus diagnostics has so far been limited to the development of an enzyme-linked oligonucleotide assay (ELONA) for apple stem pitting virus [72] but their extensive application in human virus diagnostics [73] suggest that aptamers are an under-used resource in plant virology. Their use as sensors in electrochemical platforms (aptasensors) offers an alternative, re-usable diagnostic configuration in which, together with other biorecognition elements (e.g., antibodies, nanobodies, oligonucleotides), targeted pathogen-derived biomarkers are detected in a variety of matrices [74,75,76]. In NZ to date the adoption of PoU methodologies for plant virus detection has been non-existent and yet there is expected to be a growing demand from both border quarantine officials and farmers to provide them with a means to quickly screen plant material for suspected viral disease.

Symptoms of cucumber green mottle mosaic virus (CGMMV) and other related tobamoviruses of cucurbits are often mild on the fruit surface, which creates a challenge for inspectors of fresh produce at the border. Some fruit that may be misshapen and have some yellowing or other symptoms on the skin are often observed during border inspections. The inspecting quarantine officer is left to decide whether a sample needs to be sent to the laboratory for testing. A rapid detection is essential due to the commodity’s perishable nature. At present PHEL provides virus test results for most urgent border samples within 3 days. If samples are positive by real-time PCR, in some cases (e.g., a new detection for that country), they are then retested with end-point PCR and the resulting amplicon Sanger sequenced for validation purposes. If the border staff had a sensitive PoU test on site, they could more effectively triage positive material within the hour, while waiting for laboratory confirmation.

Equally, in the event of a disease outbreak growers could more effectively monitor suspicious symptoms in their fields and greenhouses. LFIAs are available from various commercial companies (e.g., Agdia^®^ and Bioreba AG) for many economically damaging viruses, although they are not cheap (around USD5 per test) or sensitive. Companies have also begun offering vastly more sensitive RPA system-based kits but that comes at a higher cost (e.g., AmplifyRP^®^ Acceler8^®^ which costs USD31 per test). Providing growers with more inexpensive tests would assist in improving disease management [77]. An outbreak of pepino mosaic virus (PepMV) in 2021 in commercial tomato greenhouses in the Auckland area [78] has prompted research at PHEL into developing a more pre-emptive surveillance diagnostic tool for environmental RNA. This work will build from the findings of Mehle et al. [79,80] to maximise the ability to detect PepMV and other tomato infecting viruses and viroids in greenhouse irrigation water.

## 3. International Networks

PHEL scientists participate in a wide collaborative network to deliver biosecurity outcomes for NZ. In NZ, the Better Border Biosecurity (B3) partnership brings together NZ’s major plant health research providers, the Crown Research Institutes and their international collaborators, and the key end-users—MPI, the Department of Conservation, Horticulture NZ, and the Forest Owners’ Association. The aim of this partnership is to explore and deliver science solutions to reduce the entry and establishment of unwanted exotic plants, pests, and pathogens into NZ. This is achieved through three research focus areas, one of which is the development of innovative diagnostic techniques which directly supports PHEL’s diagnostic role.

PHEL also collaborates directly with many overseas quarantine laboratories involved in national plant protection work (e.g., United States Department of Agriculture [USDA]; French Agency for Food, Environmental and Occupational Health & Safety [ANSES]; United Kingdom Department for Environment, Food and Rural Affairs [DEFRA]; Australian Government Department of Agriculture, Fisheries and Forestry [DAFF]) and represents NZ in international plant protection bodies providing leadership in biosecurity diagnostics.

NZ is one of the over 180 countries signatory to the International Plant Protection Convention (IPPC), which is a multilateral treaty for international cooperation in plant protection. The IPPC develops international standards for phytosanitary measures (ISPMs) which aim to harmonise phytosanitary measures for the purpose of facilitating international trade. PHEL contributes directly to the IPPC by reviewing different ISPMs, and in the development of diagnostic protocols for inclusion into ISPM27: Diagnostic protocols for regulated pests. PHEL staff also represent NZ in the Quadrilateral Scientific Collaboration in Plant Biosecurity (Quads). This initiative was established in 2006 through the Plant Health Quadrilateral agreement between Australia, Canada, NZ, and the USA. This initiative provides a framework for scientific cooperation in dealing with phytosanitary issues and for identifying and sharing plant-protection-based tools and technologies.

MPI joined the European Network for Phytosanitary Research Coordination (Euphresco) in December 2019. This has provided an opportunity for PHEL scientists to join a number of projects, including the recently completed project on the validation of molecular diagnostic methods for the detection of tomato brown rugose fruit virus (ToBRFV) in tomato and pepper seeds (project 2019-A-327) [81].

PHEL is also involved in several collaborations in the Pacific region and with Australia. The New Zealand Aid programme ‘Enhanced Pacific Biosecurity Partnership’ involves several Pacific Island countries (Cook Islands, Fiji, Niue, Samoa, Tonga, and Vanuatu) and NZ’s MPI and Ministry of Foreign Affairs and Trade (MFAT) and aims at improving biosecurity in the Pacific. This programme is mutually beneficial as there is considerable movement of people and goods between the Pacific and NZ. PHEL scientists have been providing training in biosecurity diagnostics, including in plant virology, to quarantine officers and laboratory staff from the national plant protection organizations of Pacific countries for several years through this programme. Other examples include the Subcommittee on Plant Health Diagnostics (SPHD) with representation from the Australian state and territory governments, Plant Health Australia, Commonwealth Scientific and Industrial Research Organisation (CSIRO), and NZ’s MPI. There are five working groups in the SPHD, and PHEL has been contributing to a number of these over the years. The work includes preparedness and response to high-impact plant pathogens, as well as reviewing or validating Australia’s national diagnostic protocols (NDP) (https://www.plantbiosecuritydiagnostics.net.au/national-diagnostic-protocol-list/, accessed on 23 May 2022). PHEL is also a NZ partner on two Horticulture Innovation Australia (HIA)-funded research projects, one of which aims at improving plant industry access to new genetic material through faster and more accurate diagnostics using HTS.

Collaborative work with Australia’s biosecurity agencies has led to the start of a project based on Trans-Tasman cooperation in PEQ services—the ‘Mickleham pilot project’, named after the PEQ facility in Australia.

### The Mickleham Project—A Pilot Initiative for Sharing of Post-Entry Quarantine and Diagnostic Services between Australia and New Zealand

The Mickleham project is a bilateral initiative aimed at harmonising trans-Tasman PEQ capabilities. Both NZ and Australia have strict quarantine measures for the importation of new plant germplasm, with an overlap of regulated pests and diseases and requirements for plants to be quarantined in Biosafety level 3B and level 2 containment greenhouses, respectively. This is a unique project that will allow the two countries to complement each other’s strengths to facilitate the importation of plant germplasm, and it provides a good example of the potential for ongoing international cooperation in PEQ optimisation strategies.

Due to Australia’s present strict laws on the importation of infected positive controls, the development of sensitive molecular assays relies heavily on synthetic controls, while existing PEQ space constraints in NZ mean that greenhouse facility Biosafety level 3B bookings are backlogged until 2024. By combining activities at the Mickleham PEQ Facility in Australia and the PHEL in NZ, the project aims to overcome both these bottlenecks with integrated propagation, diagnostic, and release operations—with strawberry (*Fragaria* x *ananassa,* Duchesne) selected as a test case. The hoped-for outcome is one pathway for importation of germplasm to both countries (Figure 3).

Initial steps to harmonise activities have involved comparing import entry requirements, sharing testing protocols, and then validating molecular diagnostic tests for those regulated pathogens across both jurisdictions where there was a gap or divergence. In total, six high-risk viruses, spanning four genera (*Polerovirus, Ilarvirus, Closterovirus*, and *Crinivirus*) and three families (*Solemoviridae, Bromoviridae*, and *Closteroviridae*) were selected for validation work including strawberry polerovirus 1 (SPV1), *Fragaria chiloensis* latent virus (FCILV), strawberry criniviruses 3 and 4 (SCrV-3 and SCrV-4), strawberry chlorotic fleck virus (SCFaV), and strawberry pallidosis-associated virus (SPaV). Some of these viruses were recently added to the regulated pathogens in Australia’s import requirements for strawberry nursery stock (https://bicon.agriculture.gov.au/BiconWeb4.0, accessed on 2 June 2022). NZ’s current Import Health Standard (IHS—Nursery stock 155.02.06; [82]) does not specifically require testing for SPV-1, SCrV-3, and SCrV-4, and thus a need for evaluation and/or development of harmonised tests for these viruses was identified. A need for alignment of test methods for FCILV, SPaV, and SCFaV was also identified due to divergent methodologies and the need to achieve equivalent biosecurity outcomes. Each virus has been reported to negatively affect strawberry and is thus a potential risk for both countries [83,84,85,86,87,88,89].

Several of these viruses have been associated with strawberry decline (SD) symptoms and yield losses. Strawberry decline is associated with a complex of viruses and symptoms including stunting, reduced root development, production of small fruit, mild chlorosis and reddening, and occasional wrinkled, curled, and deformed leaves that may exhibit mottling and chlorotic spots [84,86,90]. An outbreak of the disease in California in 2002 and 2003 caused an estimated loss of more than USD50 million [91]. Recently, HTS has been playing a crucial role in unravelling the etiology of several diseases in strawberry [92], including the identification of SPV1, SCrV-3, and SCrV-4 associated with decline symptoms. SPV1 is a tentative member of the genus *Polerovirus* in the family *Solemoviridae,* which was first identified in Canada in 2013 [89] and subsequently reported in the USA in 2016 [93] and in the Czech Republic in 2021 [94]. SCrV-3 and SCrV-4 are members of the genus *Crinivirus* in the family *Closteroviridae*, which were first identified in Canada in 2016 [84] and have subsequently been reported in China in 2018 [83].

The Mickleham project has just concluded with the release of plants from PEQ in both countries in late November. Both governments are currently reviewing the results and innovations from this novel approach on sharing of PEQ and diagnostic services, with the outcomes being published upon completion. However, the collaborations established go beyond this project, with participants also working together developing and sharing HTS protocols for application in biosecurity as part of the post-entry quarantine diagnostic process. HTS alongside standard molecular techniques is playing a crucial role in understanding the aetiology of virus diseases in important agronomic crops (i.e., [95]). In strawberry alone, since 2019 an additional four viruses have been identified including strawberry virus 1 [96], strawberry-associated virus 1 [97], *Fragaria vesca*-associated virus 1 [98], and strawberry virus A [99]. The economic significance and biosecurity risks associated with these viruses are yet to be assessed. The ongoing international cooperation in PEQ optimisation strategies will certainly play a crucial role in future changes in the regulatory settings for strawberry nursery stock and the tools used for virus diagnostics.

## 4. Conclusions

The perfect scenario for virus diagnostics, as far into the future as the existing lens can take us, is for there to be a technology which would provide an instant sensing of living, dead, or environmental material for the presence or absence of a pathogen. As the diagnostician’s tools become more in silico and synthetic it will become all the more important not to lose touch with the biology, nor the more traditional tools used to arrive at this point in our understanding of virology. Due to their small size and simplicity this is more relevant for viruses and viroids than for other pathogens, where morphology still plays a prominent role in diagnostics, except, of course for liberibacters and phytoplasmas, the diagnostics of which have also benefitted from the advances of in silico diagnostic tools.

In some ways virology can be viewed as being at the forefront of diagnostics development driving ever-more-sensitive biomarker detection methods. These technological developments will inevitably result in more detections of more viruses and viroids both pathogenic and non-pathogenic, thus enhancing NZ’s surveillance capabilities. However, it is becoming more and more evident that these new discoveries cannot be viewed in isolation but should rather form part of an ongoing process to better understand the biology of viruses and the expanding virome. This will require a multi-layered strategy that draws across disciplines and agencies to ensure a responsive science-based approach to national biosecurity. This national strategy must also be framed within the principles and obligations of the Treaty of Waitangi. In an ever-changing environment, with pressures exacerbated by climate changes, international scientific cooperation will also play an essential role in dealing with emerging phytosanitary issues, and for identifying, developing, and sharing the diagnostic tools and technologies of the future.

## Figures and Tables

**Figure 1 viruses-15-00418-f001:**
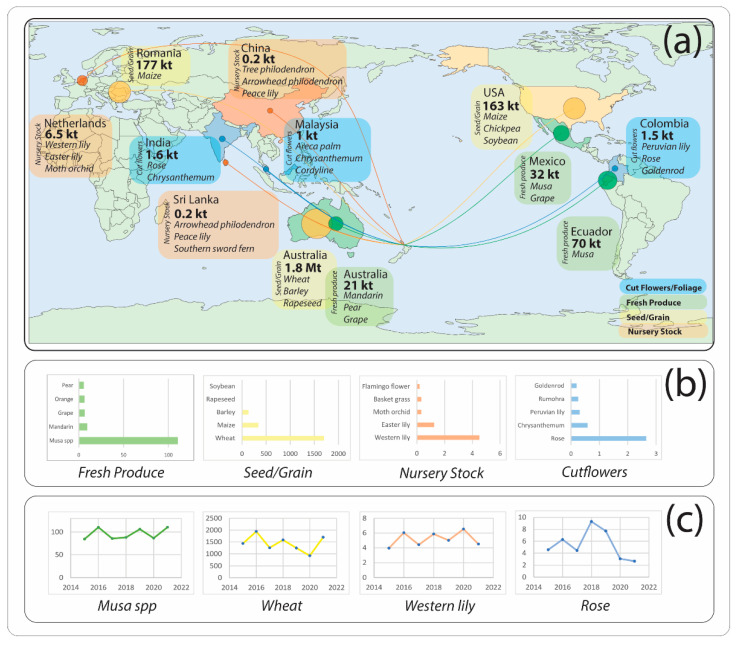
Major sources and types of imports of plant germplasm into New Zealand. (**a**) Top three country exporters by weight (t) to New Zealand per product class (Cut Flowers, Fresh Produce, Nursery stock, and Seeds/Grain) for 2021. Top three or fewer plant commodities are listed below weight (in bold). Circle size depicted relative to weight exported. (**b**) Histogram of total import weight of top five commodities (kt) for each product class (Cut Flowers, Fresh Produce, Nursery Stock, and Seeds/Grain) for 2021. (**c**) Plot of total import weight for each top commodity (kt) for each product class (Cut Flowers, Fresh Produce, Nursery Stock, and Seeds/Grain) for each year from 2015–2021. *Musa* refers to both bananas and plantains.

**Figure 2 viruses-15-00418-f002:**
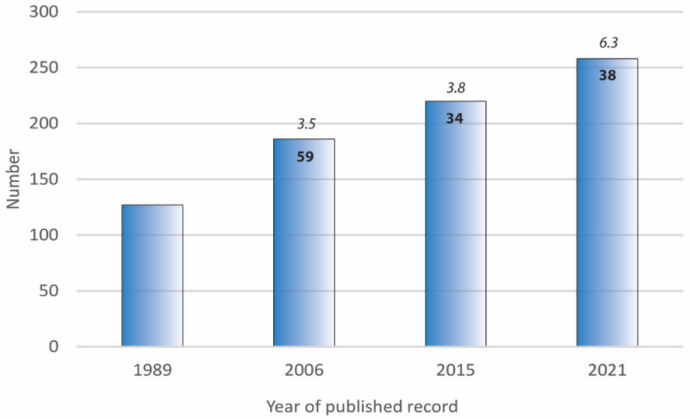
Cumulative number of plant viruses reported in New Zealand. Bold number—change from previous record. Italicized number—average change per year.

**Figure 3 viruses-15-00418-f003:**
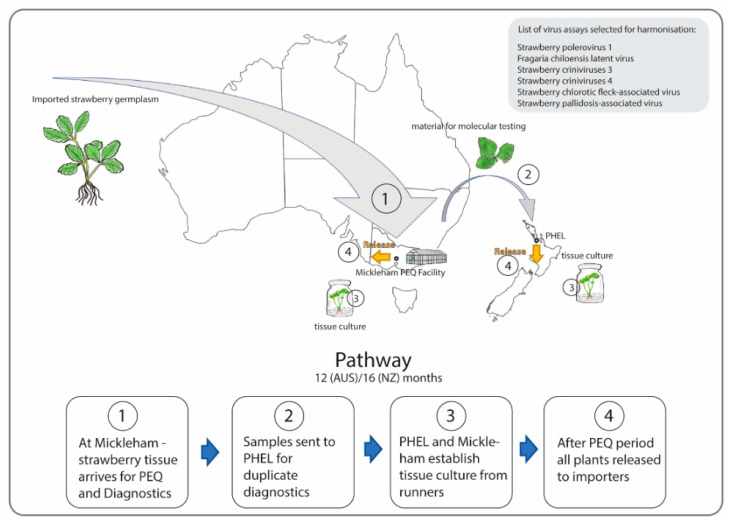
Schematic overview of each step (1–4), from importation to release, of post-entry quarantine of strawberry germplasm material within the framework of the bilateral Mickleham project.

## Data Availability

No new data were created or analyzed in this study. Data sharing is not applicable to this article.

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
