# Peer review of "New Virus Diagnostic Approaches to Ensuring the Ongoing Plant Biosecurity of Aotearoa New Zealand"

_viruses, 2023, doi:10.3390/v15020418_

Round 1
Reviewer 1 Report (Previous Reviewer 3)
I think the manuscript is ready for publishing
Author Response
Thank you very much for the positive review. We have checked and addressed minor spelling or grammar error.
Regards
Catia
Reviewer 2 Report (Previous Reviewer 2)
The review article is good in its ideas, but it needs to be checked for grammar. I have found a lot of errors, so please rewrite the review and correct the English spelling errors or fatal grammar errors, like in the abstract line 25: "has cause."
Please send us a plain document with no tracking. Furthermore, the authors must adhere to the journal's guidelines in writing their review.
Rewrite the sentence at line 69 to be more informative.
Author Response
Thank you very much for your review. The senior editor has recommended that the abstract be rewritten, thus we have done so and addressed any grammatical errors in the process. We have also reviewed the rest of the manuscript for grammar and spelling issues, and have made several other small changes to improve the manuscript as a result of other reviewers’ comments.
The sentence in line 69 was rewritten to be more informative, as follow:
“Lesser recognised crops such as avocado and hops have increased in export value in the past ten years - avocado fruit and oil exports increasing from NZ$62 million (m) in 2011 to NZ$211m in 2021, and hops from an estimated NZ$9m to NZ$36m for the same period [12].”
We are resubmitting the manuscript file in 2 versions this time - one with track changes (as suggested by the senior editor) to show the changes made to address reviewer comments, and a clean version with no track changes (pdf version). We hope this will expedite the screening of this final submission.
Regards
Catia
Reviewer 3 Report (Previous Reviewer 1)
An annotated version is provided to have a better form and appropriate wordings. The main concern is the mention of private enterprises in the text (there is a note for this in the annotated manuscript) that sound like advertisement and usually is not done in scientific manuscripts.

Author Response
Thank you very much for your review. Following the suggestion of the senior editor the abstract has been rewritten, and in the process all grammatical issues were addressed. We have also revised the rest of the manuscript for any grammar or spelling issues, and I have incorporated most of the suggested changes from the annotated version which you kindly provided.
One change that we did not incorporate was changing ‘real-time PCR’ to ‘quantitative PCR’. We purposefully used ‘real-time PCR’ because even though the technology allows for quantification of the amplified target, it is used in a qualitative manner (presence/absence) in our laboratory. Thus, in the context it was being used in the text, we believe it would be misleading referring to it as ‘quantitative PCR’.
The sentences that made references to specific commercial companies have been amended, as the companies, or products, were meant to be provided as an example, rather than an advertisement. The sentences now read:
“A number of PoU testing kits using recombinase polymerase amplification (RPA) technology both in end-point and real-time formats are now commercially available for plant pathogen diagnostics (e.g., from Agdia®).”
“LFIAs are available from various commercial companies (e.g., Agdia® and Bioreba AG) for many economically damaging viruses, although they are not cheap (around US$5 per test) or sensitive. Companies have also begun offering vastly more sensitive RPA system-based kits but that comes at a higher cost (e.g., AmplifyRP® Acceler8® which costs US$31 per test).”
In regard of italicising ‘e.g.’, while I agree that it is an abbreviation of Latin and italicising was historically common practice, in modern times this is no longer the case. I have also checked multiple articles published in ‘Viruses’ in 2022, none of which used italics in this manner. To maintain consistency with the other publications in this journal, we have not made this suggested change, but we have added the comma afterwards, which seems to be common practice in American English, and it was applied in all those 2022 articles checked.
Regards
Catia
Round 2
Reviewer 2 Report (Previous Reviewer 2)
The review article now improved and could be accepted for publication
This manuscript is a resubmission of an earlier submission. The following is a list of the peer review reports and author responses from that submission.
Round 1
Reviewer 1 Report
The manuscript is far too long and present several parts that are not related to the topic treated, it must be shortened and focused following the subject (plant viruses) and to the location (New Zealand). The manuscript just describe a general word situation and general detection methods for viruse. It mix viruses and bacteria (phytoplasmas are not viruses..).
Reviewer 2 Report
The review work is very intelligent. It well organized and mainly pointed to improve the plant virus biosecurity infrastructure and the integration and development of new virus diagnostic technologies
Really l have no comments to say to the authors except in line 166 remove the yellow highlight
I appreciate accepting it in its current form
Regards
Reviewer 3 Report
The review is very important in the field and informative, also written well. My concern about the narrative is too much if it is possible to include some data figures or tables will be better.